# Development of a tool for assessing quality of comprehensive care provided by community health workers in a community-based care programme in South Africa

Frances Griffiths,[1,2] Olukemi Babalola,[2] Celia Brown,[1] Julia de Kadt,[2] Hlologelo Malatji,[2] Margaret Thorogood,[1,2] Yu-hwei Tseng,[2] Jane Goudge[2]

[1]University of Warwick Warwick Medical School, Coventry, UK
[2]Centre for Health Policy, University of the Witwatersrand Faculty of Health Sciences, Johannesburg, South Africa

**Correspondence to**
Professor Frances Griffiths;
f.e.griffiths@warwick.ac.uk

## ABSTRACT

**Objective** To develop a tool for use by non-clinical fieldworkers for assessing the quality of care delivered by community health workers providing comprehensive care in households in low- and middle-income countries.

**Design** We determined the content of the tool using multiple sources of information, including interactions with district managers, national training manuals and an exploratory study that included observations of 70 community health workers undertaking 518 household visits collected as part of a wider study. We also reviewed relevant literature, selecting relevant domains and quality markers. To refine the tool and manual we worked with the fieldworkers who had undertaken the observations. We constructed two scores summarising key aspects of care: (1) delivering messages and actions during household visit, and (2) communicating with the household; we also collected contextual data. The fieldworkers used the tool with community health workers in a different area to test feasibility.

**Setting** South Africa, where community health workers have been brought into the public health system to address the shortage of healthcare workers and limited access to healthcare. It was embedded in an intervention study to improve quality of community health worker supervision.

**Primary and secondary outcomes** Our primary outcome was the completion of a tool and user manual.

**Results** The tool consists of four sections, completed at different stages during community health worker household visits: before setting out, at entry to a household, during the household visit and after leaving the household. Following tool refinement, we found no problems on field-testing the tool.

**Conclusions** We have developed a tool for assessing quality of care delivered by community health workers at home visits, often an unobserved part of their role. The tool was developed for evaluating an intervention but could also be used to support training and management of community health workers.

## INTRODUCTION

Many low and middle-income countries are deploying community health workers to improve access to heath care[1] by underserved communities. Much of their work involves visiting households to screen, case-find and refer and give advice on health promoting actions including treatment adherence. Some programmes include detection and treatment of acute infection[2 3] and many programmes tackle single disease areas such as screening for tuberculosis.[4] However, a growing number of countries, including South Africa, are broadening the remit of community health workers to provide comprehensive care. They work across a range of conditions, for example tracing children with missed immunisations or medication defaulters, identifying individuals with a persistent cough and monitoring patients with long-term conditions such as HIV or diabetes.[5 6] There is growing evidence that such workers can provide effective care,[7–11] although when such programmes are scaled up they can fail to produce the expected benefits due to inadequate supervision.[12]

## Strengths and limitations of this study

► The quality of community health worker household visits is difficult to assess as this is often an unobserved part of their role.
► We developed a tool that can be used by trained fieldworkers to observe community health worker household visits to assess the quality of care delivered.
► The tool was developed with input from fieldworkers who might use it.
► The tool will need further testing and development with a wider range of community health workers and recipient households, and in other low- and middle-income countries.

There is currently no consensus on the best way of assessing the quality of care provided by community health workers who are tasked with providing comprehensive care. Methods that have been used include:

► Knowledge test, monthly self-reported activity and household coverage rate.[13]
► Knowledge test plus observation by a medical officer of consultations with sick children.[14]
► Itemising the contents of community health workers' bags, assessing their ability to report their activity and a clinician observing two consultations with children.[15]
► Supervisors' global impressions.[16]

A systematic review of intervention design factors that influence performance of community health workers included 22 papers that considered comprehensive care provision[17] of which only two papers reported measurement of community health worker performance. One of them used self-report in a survey of community health workers[18] and in the other community health workers brought patients to an assessment and a clinician observed the patient consultation.[19] We have developed an observation tool for use by fieldworkers shadowing community health workers on their household visits.

Quality of care is a complex concept,[20] and any one tool can only assess a limited number of components. Measuring outcomes makes sense for programmes tackling single diseases but for comprehensive care there are many potential outcomes and any assessment focusing on only a few of these may miss important elements of care quality, while trying to measure them all would be infeasible. Following and adapting the seminal framework proposed by Donabedian[21] we have focused in this study on measures of structure (eg, health service links, equipment and logistics[22]) and process (eg, competency in communication, adherence to standards and procedures[22] and activities[23–25]). This paper describes the development of our quality of care assessment tool and accompanying training manual for assessing the quality of care provided during household visits by community health workers providing comprehensive care.

### Health system context

We developed the tool during 2016 to 2018 in South Africa where many communities have limited access to healthcare, there are shortages of professional nurses and overcrowded primary care clinics. In response to these problems, in 2011, the South African government brought community health workers into the public health system. Teams of community health workers are now attached to local primary care clinics to provide comprehensive outreach health promotion, prevention and screening. Their standardised training covers identification of the need for antenatal and postnatal care, monitoring immunisation of under 5 s, adherence among patients with chronic diseases, screening for malnutrition, tuberculosis (TB), gender-based violence and making referrals to health, social and other services. They are not trained in detection of acute infection. Many of them had

previously been employed by non-governmental organisations focused on single health issues such as HIV or home-based care. Most of them had minimal training and some were illiterate. During 2016 to 2018 there was political unrest among the community health workers because of unwelcome changes to their contracts and payment system.

### METHODS

We developed the quality of care assessment tool for use by non-clinical fieldworkers observing household visits conducted by community health workers providing comprehensive care. We were guided by Van Der Vleuten's assessment utility model considering the validity, reliability, feasibility and acceptability[26] of the tool.

### Study context

Our development of the assessment tool formed part of a larger study in which we set out to improve the quality of the care provided by community health workers in one health district of Gauteng Province in South Africa through improving supervision and training.[27] Our plan was to use the tool to assess quality of care delivered before and after the intervention. Prior to developing and delivering the intervention we qualitatively explored how six community health worker teams functioned.[12] The teams were based at different primary care centres and served socioeconomically deprived populations. Our qualitative data included fieldworker observation of community health workers, working in pairs, undertaking home visits over a total of 126 days. Community health workers were randomly selected and once selected were observed for 3 to 5 days. The fieldworkers took brief notes during observation and then expanded them to include place, people, activities and interactions plus the fieldworker's impressions.[28] We observed 70 of the 88 available community health workers. None of them refused to be observed and all 518 householders visited during observations allowed the fieldworker into the household. We used the data in the development of our tool.

### Patient and public involvement

The need for the larger study, of which this formed part, was identified in collaboration with the health district manager. We involved district, clinic and community health worker programme managers in the design of the study. We did not involve patient and public in this study. We plan to include householders in further development of the quality of care assessment tool (see discussion).

### Why we chose our approach to assessing quality of care

We focused on household visits as these take up the majority of community health workers' work-time and typically require independent work. From our exploratory study we knew households were likely to have many different health and social needs. We considered and rejected using formal knowledge tests for the community

**Table 1** Components of the quality of care tool and the information sources used to generate content

| Tool sections (point at which recorded) | Components of the tool | Source of data for component, item and score generation |
|---|---|---|
| Before setting out | Contents of CHW bag | Interaction with district managers |
| Just before and on entry to a household | Visit planning | Observations of CHW undertaking household visits during exploratory study |
| | CHW communication skills including attention to confidentiality | South African national training manuals* for CHW[32 33]; Published frameworks and tools for assessing health professional communication skills |
| During household visit | Householder conditions and messages and actions expected of CHW | Interaction with district managers; South African national training manuals* for CHW[32 33] |
| | CHW communication skills including attention to confidentiality | South African national training manuals* for CHW[32 33]; Published frameworks and tools for assessing health professional skills |
| After leaving the household | Factors that would prevent a CHW delivering good quality care | Observations of CHW undertaking household visits |
| | CHW communication skills including attention to confidentiality | South African national training manuals* for CHW[32 33]; Published frameworks and tools for assessing health professional communication skills |

*Used for training of CHW in the study health district; Phase 1 and Phase 2 training both comprise 10 days classroom-based plus observed and assessed household visits.
CHW, community health worker.

health workers as this would only assess what they could do and not what they actually do. We considered and rejected using routinely collected activity data as data quality can be poor,[25] and activity counts do not provide evidence of activity quality. We also considered but rejected using a nurse as observer. This option is expensive, nurses are in short supply and community health workers are likely to defer to the nurse. Householders are likely to expect the nurse to use their knowledge and skills to assist them, not to be a passive observer and the nurse might feel obliged to intervene. We had found that fieldworker observation was acceptable to community health workers and householders, so we decided on an assessment tool for use by non-clinical fieldworkers. A similar approach has been used successfully to assess quality of care in medical consultations.[29–31]

### The structure of the tool

We structured the sections of the tool to follow the flow of the working day for a community health worker conducting household visits: before setting out, then for each household visit - just before and on entry, during household visit and after leaving the household (table 1). This provides a convenient order for tool completion by the fieldworker. Most items require a categorical response (eg, present/absent).

### Sources of information used to develop components of the tool

To ensure the content validity, we used multiple sources of information to generate the tool components, their constituent items, the specific actions being assessed and scores for each item or group of related items. Table 1 shows tool components and information sources.

### How we generated component items and scores
#### Sources of information
*From the national training manuals for community health workers*
Two members of the team independently read the manuals and listed the health conditions to be addressed, the actions to be taken and messages delivered during household visits.[32 33] We included communication skills although omitted ones such as communicating with a child where abuse is suspected, as this skill is unlikely to be used in the presence of a fieldworker. The resulting lists were combined and discussed.

*Using data from the district managers*
We collated lists of equipment issued for household visits and, activities community health workers were expected to undertake that were not included in the national training manual, such as delivering medication to older people.

*From our observations of community health workers undertaking household visits*

Members of the team read and re-read transcripts of the observations of community health workers' household visits from the exploratory study. The author team then met to brainstorm what to put in the tool. After developing an initial list, we re-read the observation data to identify missing items, continuing until we were finding no new items.

*Using published frameworks for assessing health professional communication skills*

We wanted to include markers of communication quality, so we searched published literature to identify key frameworks of assessment of health professional communication.[34–37] We extracted and listed assessment domains such as rapport building and involving patient in planning healthcare. We then considered which were within the remit of community health workers and identified markers of quality. For example, did the community health worker use their previous knowledge of the household to ask questions, did they interrupt the patient, did they attend to privacy?

### Developing quality of care outcome scores for intervention study

For our planned evaluation we wanted scores that summarised key aspects of the care provided by community health workers. Other data collected using the tool would provide contextual information to complement the scores. We developed scores on (a) delivering messages and actions during the household visit and (b) communication with the household.

*Score for messages and actions delivered by the community health worker during the household visit*

We developed a list of health states and the relevant actions and messages expected of a community health worker, for example, giving advice on diet, exercise and medication adherence for someone with diabetes, checking a child's parent-held immunisation record, asking women about family planning needs, asking about cough. We excluded actions that the community health worker might not attempt in the presence of a fieldworker such as identifying abuse. We then developed a method of scoring messages and actions which takes account of each householder's health needs. For each condition, we identified the expected messages and actions. For example, for hypertension there were four expected messages and actions: (a) asking and advising about food/exercise, (b) asking about medication adherence/side-effects, (c) measuring blood pressure and (d) checking access to medication supplies. If a householder had a condition such as hypertension, the community health worker was scored on the number of messages or actions delivered. If two people in a household had a condition that requires the same message and the message is delivered to them both at the same time, this was recorded as two messages. If a householder required the same message or action for more than one condition, these were recorded separately for each condition. We calculated the proportion of expected messages and actions that were undertaken for each household.

*Quality of communication score*

We limited our list of items for assessing communication to items that could be given a categorical response and where assessment would not require extensive training of the assessor. For example, whether the community health worker interrupted the householder when the householder first started to talk about themselves. Each item was scored as 1/0 (yes/no). We calculated the proportion of achievable score for the household. Author HM had been a member of the fieldwork team undertaking observation of household visits. With HM we reviewed notes of eight household visits to consider the feasibility of assessing communication skills.

We assessed face validity and qualitatively assessed inter-rater reliability for both scores using our observation data. Randomly selected household visit observations were read and scored independently by at least two people. Scores were compared, and discrepancies discussed. Problems with scoring were identified and resolved. This process continued until scoring was consistent across scorers and no new issues arose – achieved after reading 40 observations of household visits for message and actions score and 29 for communication score.

### Development of manual for fieldworkers

From our exploratory study we identified various types and content of community health worker visits and developed guidance on how to complete the quality of care tool. For example, what to do about visitors to the household that engage with the health worker about their own health; what to do about people who are not present during a household registration and who are well or those not present who should be receiving attention from the health worker.

### Testing the tool and training the fieldworkers

To refine the tool and manual we worked over 6 days with three fieldworkers who had previously undertaken household observations, studying a further 73 household visit observations. Initially, the fieldworkers familiarised themselves with the assessment tool and draft manual, applying it to the notes of household observations. Problems identified were discussed and revisions made, for example adding response options to items. We then used role play where the research team played the community health workers and householders and the fieldworkers independently completed the tool. The fieldworkers compared and discussed results and the process continued until consistency was achieved. The format of the paper-based tool was refined to make recording as easy as possible while standing observing a household

**Table 2** Sections of quality of care tool and the information collected

| Tool sections | | Data collected |
|---|---|---|
| 1 | Before setting out | About the CHW: Site, age, education, CHW training, length of service<br>Contents of CHW bag on that day |
| 2 | Just before and on entry to a household | Unique IDs for fieldworker, CHW, household visit and patient<br>When last visited this household<br>How often normally visit<br>Plan for the visit<br>Description of dwelling<br>GPS coordinates<br>Start time of visit<br>Where did the visit take place (inside/outside)<br>Initial introduction by CHW and communication between CHW and householder |
| 3 | During household visit | Age and gender of household members<br>Health conditions, and health needs identified by CHW<br>Advice and messages given by CHW<br>Type of referral if given<br>CHW's plan for next steps<br>Whether patient engaged in making the plans |
| 4 | After leaving the household | End time of the visit<br>Any communication difficulties between CHW and patient<br>CHW's sensitivity to privacy<br>Did the CHW make notes<br>Any problems with the consultation (disruptions, negative attitudes from household) or barriers to ensuring patients' access care<br>FWs assessment score of CHW visit<br>CHWs own assessment score of the CHW visit |

CHW, community health worker; FWs, fieldworkers; GPS, global positioning system.

visit, often in the confined space of a living room of a shack or small house. We digitised the tool for data entry. However, during a household visit the community health worker's (CHW) attention may switch between household members. It was important for the fieldworker to be able to switch quickly between different sections of the tool. With the fieldworkers we decided this was easier to do on paper than on the small screen size of the smart phones available to us. Once the tool and manual were finalised, the fieldworkers continued to practice data collection using role play until they were able to complete the assessment in real time and produce consistent assessments. Finally, the fieldworkers used the tool at a different site from our study sites and provided feedback on feasibility.

## RESULTS

The tool consists of four sections, each completed at different stages during household visits. Table 2 shows the data collected at each stage. For each of these stages we describe the section content and reasons for including the various items.

### Before setting out for household visits

This is a list of the equipment carried by the community health worker on home visits. Fieldworkers asked if equipment was in working order as poor or missing equipment impacts on the quality of care.

### Just before and on entry to a household

The fieldworker asks the community health worker about their previous engagement with the household and their plans for this visit, records the global positioning system (GPS) coordinates and start time of visit, then observes how the community health worker initiates the visit. There is a section for recording whether the householder agreed to the fieldworker observing the community health worker doing her work and, for households where a planned visit was not undertaken, the reason for this, for example, the householder did not have time or there was no one over the age of 18 present at the time of the visit. Also included in this section are items that contribute to the communication score.

We asked about previous encounters with the household as this is likely to influence communication and visit content. For example, some households with older people are visited every month to deliver medication and for other households a household registration visit might be the first time the householder has encountered the community health worker.

There is usually a plan for each household visit, for example follow-up of a householder discharged from hospital, checking on a frail elder who lives alone or defaulter tracing.

Community health workers are not always welcomed into households so sometimes interactions are carried

**Table 3** Additional health needs added to the tool during development

| | Health need | Examples |
|---|---|---|
| 1 | Routine checks | Checking on frail old man, advising about family planning, asking about cough, asking about social grants. |
| 2 | Other illness or potential illness | Householder asking for advice on treating an injured hand or on a child's rotten teeth. |
| 3 | Other or unknown chronic illness | Householder has asthma and talks about access to medication. The nature of the illness is not mentioned during the visit because it is HIV which carries stigma, or because the health worker and householder know each other well so there is no need to mention the condition. |

out at the yard gate or household door. This limits what actions the community health worker can undertake and the quality of communication.

We asked for time of starting and finishing the visit because there is evidence that longer service delivery time is associated with higher health worker performance.[38] During our exploratory study we found the average duration of household visits was 16 min for teams with the highest levels of training and a professional nurse support, and 11 min for teams with lowest level of training and no professional nurse support.

GPS coordinates allows location of households visited to be compared with available maps of the locality to reveal gaps in location coverage.

### During the household visit

While the community health worker carries out the visit, the fieldworker focuses on recording details of all household members present, including their health needs and actions taken by the community health worker. Based on training manuals and interaction with district managers, the list of health needs comprised: children under 5 years old, pregnant women, individuals with persistent cough and those known to have HIV, TB, diabetes or hypertension. When we tested the draft tool on observation data, we found and added three further categories of health needs (table 3).

For each individual with a health need identified by the community health worker, the fieldworker marks the messages and actions delivered. If the community health worker took the initiative to find out about further health needs of household members or to check-up on a household member who they already knew, this was captured within routine checks. We did not assess response to the needs of household members who were not present during the visit, nor of visitors to the household. We recorded plans made such as the worker returning for another visit, facilitating access to care at the clinic by speaking to the nurse, speaking to a non-governmental organisation about food parcels or facilitating access to the state agency dealing with benefit payments.

When a referral was made, for example to attend a clinic, we recorded how it was made, such as verbally, on formal referral form or written on scrap of paper. In our exploratory study we found community health workers often do not have copies of the formal referral form.

### After leaving the household

In this section of the tool we included items that are important for explaining and understanding the quality of care. These include problems with communication with householders, challenges in undertaking the visit such as disruptions and unresolved barriers to householders gaining access to care. We also recorded whether the health worker made notes about the visit.

We had observed visits where the community health worker was unable to communicate with the householder due to deafness, cognitive disability or lack of a shared language. The latter problem was more common in informal settlements with incoming migration. We also observed challenges such as visitors walking in during a visit, disruption of the visit because the householder or visitors were intoxicated and uncooperative householders. Barriers to gaining access to care mentioned by householders included lack of transport, medication shortages and dismissive staff.

### Overall review of the household visit

We collected two overall assessments of the visit as the community health worker and fieldworker left the household. The fieldworkers were instructed to score (scale 1 to 5) how well the CHW performed given the circumstances, for example, taking time to talk through a patient's concerns about transferring to a different clinic would be given a high score. They were not asked to judge clinical quality. The fieldworker then asked the CHW to give an overall score (scale 1 to 5) on how happy they were with how the visit went.

When we tested the feasibility of using the tool in the field we found no problems.

### DISCUSSION

Assessing the performance of community health workers during household visits is challenging but critical for improving quality of care provision. We have developed a quality of care tool to evaluate performance during household visits using observation by non-clinical fieldworkers for use in an intervention study. Previous assessments

of community health worker quality of care have been undertaken using observation[14][15][19][39][40] but not within the normal work setting of households in the community. Our tool assesses an aspect of community health worker activity that is mostly undertaken unsupervised and so not often evaluated. The content details of the tool can be adapted to local community health worker programme expectations in other settings.

Using the tool, data are collected for two scores – messages and actions and quality of communication – for use as outcome measures in our intervention study. However, it is essential to contextualise these scores using other data captured in the tool. For example, scores should be reported along with data on disruptions during household visits. The tool provides extensive process data, for example the number of relevant referrals made, along with contextual data such as the number of households refusing to go to the clinic because of their previous bad experiences.

The tool does not assess quality of care as perceived by the householder, although this can influence whether people seek and accept care.[20] This would demand an independent visit to the household. The health worker may make an effort to improve their performance when observed, although this effect tends to wear off as observation continues.[41]

Our tool covers aspects of pre-service training described in the 2018 WHO guideline on optimising community health worker programmes. These include health promotion, identifying the health and social needs of households, referral to clinics or other agencies and communication skills.[42] The guideline conditionally recommends certification of competency after pre-service training but acknowledges there is insufficient evidence of effectiveness. It is silent on the assessment of community health workers while in-service.

### Strengths and limitations of the development process of the quality of care tool

The tool is based on current expectations of community health workers and extensive observation of community health worker household visits to ensure it has face validity. However, the communication assessment frameworks used to inform the communication score were developed for doctors or nurses rather than community health workers. Our check for reliability was undertaken iteratively while developing and refining the tool, so further reliability testing in the field is needed. Although we know that householders accepted a fieldworker observing a community health worker visit, we have not formally asked householders and health workers for their views on the acceptability of using the quality of care tool.

The tool is undergoing further testing as part of the larger study. This includes evaluating face validity with community health workers, their supervisors and householders, measuring inter-rater reliability in the field and assessing sensitivity to change. We will assess concurrent validity by comparing the duration of household visit to assessment scores, as there is evidence that higher care quality is delivered when visits are longer.[38] In developing the tool, we assumed that there is one underlying competency called 'quality of community health worker care' which is captured consistently across observations. To test this, using the global assessment data from both the field-worker and the community health worker we will determine which tool components contribute the most to this overall judgement. The tool needs further testing for use with community health workers in other contexts.

Further research is needed to ensure that the tool reliably assesses individual health worker performance. The sampling process for household visit assessment using the tool will need to ensure a range of visit types and purposes are observed for each community health worker as context may unduly influence performance ratings, even when we take into account the contextual issues captured in the tool. The consequence would be that the number of observations required to obtain a reliable estimate of the quality of care provided by any one CHW could be fairly large.[43]

Where suitable devices are available, the digitised version of the tool may facilitate its further use.

### Potential use and development of the tool for informing in-service development of community health workers

Our tool was developed with the evaluation of an intervention in mind. However, our tool could also be used to understand and plan for education and development needs.[26] Monitoring and evaluation of community health worker programmes is one aspect of providing strong governance of these programmes.[44] Quality assessments made through observation can form part of supportive supervision and contribute to improved healthcare provision[45] although the 2018 WHO guideline on health policy and system support to optimise community health worker programmes indicates the evidence for the use of feedback based on performance data is limited.[42] Our tool provides a structure for the observation of household visits undertaken by community health workers that could be used to provide them with purposeful and effective feedback.

**Acknowledgements** We thank the community health workers and their team leaders, health facility managers, Provincial and District health teams of Gauteng Province and local government councils. We thank the data collection team members: Nompumelelo Ngcobo, Nthabiseng Mofokeng, Aubrey Mantsho, Andani Singo, Maletsatsi Matona, Nhlovo Tsuvuka, Nompumelelo Mbanjwa, Hlokoma Mangqalaza and Nahledi Mahlangu.

**Contributors** JG, FG and Thorogood designed the research and supervised the data collection used to inform development of the assessment tool. HM and JdK supervised and undertook data collection. FG, Goudge, MT, HM, JdK, OB and Y-hT contributed to data analysis. FG led the tool development process and JG, MT, HM, JdK, OB and CB contributed. FG drafted the manuscript and JG, MT, HM, JdK, OB, CB and Y-hT contributed substantially to the paper.

**Funding** This work was funded by Medical Research Council grant number MR/N015908/1.

**Competing interests** None declared.

**Patient consent for publication** Written consent was obtained from all participants.

**Ethics approval** Human Research Ethics Committee (Medical), University of the Witwatersrand M160354 Biomedical and Scientific Research Ethics Sub-Committee, University of Warwick REGO-2016-1825.

**Provenance and peer review** Not commissioned; externally peer reviewed.

**Data availability statement** No additional data are available.

**Open access** This is an open access article distributed in accordance with the Creative Commons Attribution 4.0 Unported (CC BY 4.0) license, which permits others to copy, redistribute, remix, transform and build upon this work for any purpose, provided the original work is properly cited, a link to the licence is given, and indication of whether changes were made. See: https://creativecommons.org/licenses/by/4.0/.

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
