## [Reviewer comments · BMJ Open]

ARTICLE DETAILS

TITLE (PROVISIONAL)	Development of a tool for assessing quality of comprehensive care provided by community health workers in a Community-Based Care programme in South Africa
AUTHORS	Griffiths, Frances; Babalola, Olukemi; Brown, Celia; de Kadt, Julia; Malatji, Hlologelo; Thorogood, Margaret; Tseng, Yu-hwei; Goudge, Jane

VERSION 1 – REVIEW

REVIEWER	Maryellen Potts, PhD University of Kansas, School of Nursing Kansas, United States
REVIEW RETURNED	19-Apr-2019

GENERAL COMMENTS	Dear Authors, Thank you for your good work on describing the development of the assessment tool for determining quality of care delivered by CHWs. You make it very clear the steps taken to develop the instrument and steps still yet to be taken to ensure its reliability and broader use outside of your intervention. I agree about the potential use for the tool regarding training and also in other low resource countries utilizing CHWs to deliver home-based care. I appreciate the clarity and completeness of description of the tool development process. Your article will make a valuable contribution to the field.
---

REVIEWER	Professor Tanya Doherty South African Medical Research Council I have been involved as part of an advisory group for the larger intervention study.
REVIEW RETURNED	02-May-2019

GENERAL COMMENTS	Thank you for the opportunity to review this article describing the development of a tool for assessing the quality of care delivered by community health workers (CHWs) in South Africa. This article makes a very important contribution to the literature as many countries are scaling up CHW programmes and supervision has been described as one of the weakest components of these programmes. The article describes the very thorough, methodologically sound way in which the tool was developed using information from document reviews and observations of CHW visits. Perhaps the authors could provide some more background information on how long CHWs in Gauteng are trained for initially and whether this is all classroom based or includes practical training with clients/ patients. Could the authors describe
--

	whether the tool is completed on paper or through an electronic medium. If it is on paper how feasible would it be to convert it to a digital application? Although this tool was developed for the assessment of an intervention study, it has potential to be applied to routine supervision of CHWs and its impact would then depend on corrective actions taken to respond to any deficiencies found in the quality of care. There are minor editorial corrections required in language.
--	--

REVIEWER	Lívia Pimenta Bonifácio Universidade Federal de Uberlândia - Federal University of Uberlândia Minas Gerais, Brazil
REVIEW RETURNED	05-Jul-2019

GENERAL COMMENTS	I would like to thank you for inviting me to review this article, since it addresses a topic in which I am very interested - community health workers and development of tools to evaluate the quality of the service and improve health care. And I congratulate the authors for the study. It is an area of extreme importance, including being curious about the larger study in which this article is involved, I would like to receive more information if it is possible. I think the article is good, but it needs minor adjustments. First contextualize more about the work situation of community health workers in the introduction. We know that in every region of the world there is a difference in the way this professional works. There are sentences in the text that describe some limitations of the study, I think they need to be described in the specific part for this (final part of the discussion). Describe more about the assembly of the tool, the items described in the table. A little more detailed information is missing, in the text it only describes that certain manuals and documents were used to produce the tool, but not HOW. Delving deeper into the design, study method for replication to be possible. There was a lack of further discussion to give strength to the study. And I was interested in seeing the final result, the tool itself, how was the instrument that was produced to be able to adjust and use in the practical routine. More information I add in the text itself with some "highlights" and "balloons" in the good parts and in those that need to change (text attached). The reviewer provided a marked copy with additional comments. Please contact the publisher for full details.
---

VERSION 1 – AUTHOR RESPONSE

Reviewer: 1

Reviewer Name: Maryellen Potts, PhD

Institution and Country: University of Kansas, School of Nursing, Kansas, United States

Please state any competing interests or state 'None declared': None declared

Please leave your comments for the authors below

Dear Authors,

Thank you for your good work on describing the development of the assessment tool for determining quality of care delivered by CHWs. You make it very clear the steps taken to develop the instrument and steps still yet to be taken to ensure its reliability and broader use outside of your intervention. I agree about the potential use for the tool regarding training and also in other low resource countries utilizing CHWs to deliver home-based care. I appreciate the clarity and completeness of description of the tool development process. Your article will make a valuable contribution to the field.

Response: Thank you for your feedback. We note your comments on the clarity of our description of the steps we took to develop the tool and the completeness of the description.

Reviewer: 2

Reviewer Name: Professor Tanya Doherty

Institution and Country: South African Medical Research Council

Please state any competing interests or state 'None declared': I have been involved as part of an advisory group for the larger intervention study.

Please leave your comments for the authors below

Thank you for the opportunity to review this article describing the development of a tool for assessing the quality of care delivered by community health workers (CHWs) in South Africa. This article makes a very important contribution to the literature as many countries are scaling up CHW programmes and supervision has been described as one of the weakest components of these programmes. The article describes the very thorough, methodologically sound way in which the tool was developed using information from document reviews and observations of CHW visits.

Response: Thank you for your feedback. We note your comments on the thorough and methodologically sound way in which the tool was developed.

Perhaps the authors could provide some more background information on how long CHWs in Gauteng are trained for initially and whether this is all classroom based or includes practical training with clients/ patients.

Response: We have added a note in Table 1 to clarify this

Could the authors describe whether the tool is completed on paper or through an electronic medium. If it is on paper how feasible would it be to convert it to a digital application?

Response: In the final paragraph of the methods we have clarified that we digitised the tool but with the fieldworkers decided that the paper version was easier to use than the digital version on the smart phones we had available. We have added a sentence in the penultimate section of the discussion about potential to use digital version..

Although this tool was developed for the assessment of an intervention study, it has potential to be applied to routine supervision of CHWs and its impact would then depend on corrective actions taken to respond to any deficiencies found in the quality of care.

Response: We agree. We have checked the final paragraph in our paper and note it conveys the same sense.

There are minor editorial corrections required in language.

Response: we have checked the manuscript and corrected some minor errors

Reviewer: 3

Reviewer Name: Lívia Pimenta Bonifácio

Institution and Country: Universidade Federal de Uberlândia - Federal University of Uberlândia, Minas Gerais, Brazil

Please state any competing interests or state 'None declared': None declared

Please leave your comments for the authors below

I would like to thank you for inviting me to review this article, since it addresses a topic in which I am very interested - community health workers and development of tools to evaluate the quality of the service and improve health care.

And I congratulate the authors for the study. It is an area of extreme importance, including being curious about the larger study in which this article is involved, I would like to receive more information if it is possible.

Response: Thank you, we would welcome direct contact from the reviewer for more information.

I think the article is good, but it needs minor adjustments. First contextualize more about the work situation of community health workers in the introduction. We know that in every region of the world there is a difference in the way this professional works.

Response: In our paragraph on Health System Context we have add a more detailed description of the role of CHWs.

There are sentences in the text that describe some limitations of the study, I think they need to be described in the specific part for this (final part of the discussion).

Response: the reviewer provides details of these instances in the comments on the pdf of the paper which we address later.

Describe more about the assembly of the tool, the items described in the table. A little more detailed information is missing, in the text it only describes that certain manuals and documents were used to produce the tool, but not HOW. Delving deeper into the design, study method for replication to be possible.

Response: The reviewer has noted the lack of detail about the role of the CHWs, which we have now added. This now provides greater clarity for the reader about what the tool was designed to assess. This in turn makes it easier for the reader to understand the steps we took in the method. We have re-read the methods and we think all the necessary detail is there, and we note the comments from Reviewer 1 and Reviewer 2 on detail and thoroughness.

There was a lack of further discussion to give strength to the study.

Response: We have reviewed our discussion section carefully. We have been cautious in our discussion as further testing of the tool is needed. We have made this clear in the discussion, as noted by Reviewer 1. We look forward to publishing further work in the future, both from our empirical study and our tool development where we can include further discussion.

And I was interested in seeing the final result, the tool itself, how was the instrument that was produced to be able to adjust and use in the practical routine.

Response: We would welcome you contacting us for a copy of the tool, as detailed at the end of the paper.

More information I add in the text itself with some "highlights" and "balloons" in the good parts and in those that need to change (text attached).

Response: Thank you for the further comments in the pdf. There were many "highlights" with no comments near them, so we have not commented on these. Below we provide the text of the reviewers comment in "balloons" and our response.

Position in paper: abstract

Comment: Informative abstract

Response: Thank you

Position in paper: Strengths and limitations bullet point 1.

Comment: A very important part to include in the text. But it did not focus on the main limitations

Response: In the section entitled "Why we chose our approach to assessing quality of care" we point out to the reader that the assessment tool is for household visits where CHWs typically work independently.

Position in paper: Second paragraph of introduction

Comment: I suggest that in the introduction you briefly describe the working context of the community health worker in this region. If there are public policies or programs that guide the work of this health professional and what they are, and to reinforce the importance of these evaluation studies in the quality of the service to guide health care

Response: We have addressed context in terms of public policy and programmes in the section on Health System Context which comes a little later in the paper. We feel taken overall the paper is clear about the importance of evaluating service quality.

Position in paper: 4th paragraph of introduction

Comment: Important this definition and rationale

Response: Thank you

Position in paper: Final sentence of 4th paragraph of introduction

Comment: Limitations

Response: We agree this is a limitation and have moved it to the limitations section in the discussion.

Position in paper: Final two sentences of section on Health System Context

Comment: possible limitation

Response: we agree, but this is a limitation of the qualitative work described under study context, so it fits in the introduction.

Position in paper: Study Context

Comment: Thinking about the bias of observation may be a limitation. The observed tends to perform a better work due to being observed.

Response: We agree and we comment on this in the discussion section of the paper. However, it is not a limitation of this study which reports the process of producing the tool.

Position in paper: Why we chose our approach to assessing quality of care

Comment: good. To base the choice

Response: Thank you

Position in paper: Paragraph entitled: iii) From our observations of community health workers undertaking household visits

Comment: I question whether this is the best method to evaluate observation data and to include in the tool. Have you had previous studies using this model?

Response: The text describes a standard approach to analysis of qualitative data: immersion in the data, crystallisation through discussion followed by re-interrogation of the data.

Position in paper: Sixth line of paragraph entitled: a) Score for messages and actions delivered by the community health worker during the household visit.

Comment: Mistake?

Response: We have corrected “we the developed” to “we then developed”

Position in paper: next to text that reads “Community health workers are not always welcomed into households so sometimes interactions are carried out at the yard gate or household door. This limits what actions the community health worker can undertake and the quality of communication.”

Comment: may also be a limitation of the study

Response: Not being welcome limits what the CHW can undertake and this is captured by the tool, but this is not a limitation of our study which is about the development of the tool.

Position in paper: Reference at the end of the sentence that reads: “We asked for time of starting and finishing the visit because there is evidence that longer service delivery time is associated with higher health worker performance³⁸”

Comment: I suggest putting in the discussion to support your finding

Response: We realise it is unusual to have references in the results. However, in this study the result is the content of the tool and why it was included. We therefore think the reference is best placed here.

Position in paper: During the household visit

Comment: important: show the main events to be evaluated, good

Response: Thank you

Position in paper: First paragraph of discussion

Comment: these are a strength of study

Response: Thank you

Position in paper: First sentence of third paragraph of discussion

Comment: a good factor to be considered in the study

Response: Thank you

Position in paper: Last sentence of third paragraph of discussion

Comment: a limitation

Response: We agree this is a limitation of observing health workers for assessing quality. However, it is not a limitation of this study which reports the process of producing the tool.

Position in paper: Last sentence of first paragraph of section on “Strengths and limitations of the development process of the quality of care tool”

Comment: an important limitation

Response: Thank you, we agree

VERSION 2 – REVIEW

REVIEWER	Professor Tanya Doherty Health Systems Research Unit, South African Medical Research Council, South Africa member of an advisory group for the larger intervention study
REVIEW RETURNED	26-Jul-2019

GENERAL COMMENTS	I am satisfied with the revisions and responses to the reviewers. I feel the paper should be accepted.
--

REVIEWER	Lívia Pimenta Bonifácio Universidade Federal de Uberlândia - Federal University of Uberlândia - Minas Gerais - Brazil
REVIEW RETURNED	23-Jul-2019

GENERAL COMMENTS	I would like to congratulate the authors again for their work and thank the editors for inviting them to review, I hope you contributed to the suggestions from the first review. I imagine it is difficult to find similar articles available in the scientific literature to support the discussion of the authors. And although I consider that scientific consistency is still lacking in the discussion of the article, I believe it is appropriate for publication. I want to contact the authors for more information about the instrument to apply in my practice. Thank you so much for the pleasure of this reading.
---